# Give Me the Facts! A Survey on Factual Knowledge Probing in Pre-trained Language Models

**Paul Youssef**[1,3] **Osman Alperen Koraş**[1] **Meijie Li**[1] **Jörg Schlötterer**[1,2,3] **Christin Seifert**[1,3]

[1]Institute for AI in Medicine (IKIM), University Hospital Essen, University of Duisburg-Essen
[2]University of Mannheim [3]University of Marburg
{paul.youssef, joerg.schloetterer, christin.seifert}@uni-marburg.de
{osman.koras, meijie.li}@uni-due.de

## Abstract

Pre-trained Language Models (PLMs) are trained on vast unlabeled data, rich in world knowledge. This fact has sparked the interest of the community in quantifying the amount of factual knowledge present in PLMs, as this explains their performance on downstream tasks, and potentially justifies their use as knowledge bases. In this work, we survey methods and datasets that are used to probe PLMs for factual knowledge. Our contributions are: (1) We propose a categorization scheme for factual probing methods that is based on how their inputs, outputs and the probed PLMs are adapted; (2) We provide an overview of the datasets used for factual probing; (3) We synthesize insights about knowledge retention and prompt optimization in PLMs, analyze obstacles to adopting PLMs as knowledge bases and outline directions for future work.

## 1 Introduction

Pre-trained language models have been a game changer in NLP. Their reliance on large unlabeled corpora for pre-training and the availability of computational resources have enabled a speedy scaling of these models. This scaling has been reflected on the performance of numerous downstream tasks in NLP (Devlin et al., 2019; Chowdhery et al., 2022; Touvron et al., 2023), and led to the wide adaptation of the *pre-train then finetune* framework.

The success of PLMs is attributed to the rich representations and the knowledge captured from the pre-training corpora (De Cao et al., 2021; Han et al., 2021; Ye et al., 2022). There has, therefore, been a huge interest in investigating and quantifying the type and amount of knowledge present in PLMs, e.g., (Davison et al., 2019; Jawahar et al., 2019; Petroni et al., 2019; Tenney et al., 2019; Roberts et al., 2020), in order to have a better understanding about which kinds of knowledge are internalized during pre-training, and to develop methods to

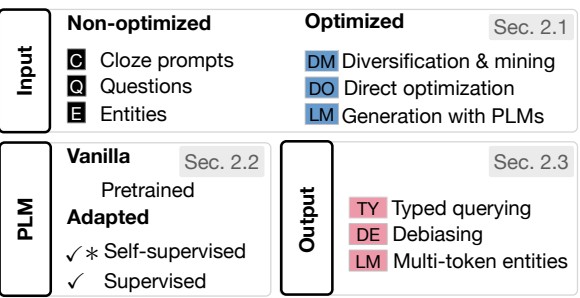

Figure 1: An overview of our categorization scheme of factual knowledge probing methods.

make PLMs more knowledge-rich and obtain gains on various downstream tasks.

Besides the interest in quantifying knowledge for better downstream tasks performance, there is a special interest in factual knowledge present in PLMs, because they are envisioned to become *soft knowledge bases*, from which one can easily extract relational knowledge that had been captured during pre-training (Petroni et al., 2019; Sung et al., 2021). Querying PLMs for knowledge would eliminate the complex NLP pipelines used for knowledge extraction, the need for labeled data to train models for relational knowledge extraction, and schema designing (Petroni et al., 2019). Furthermore, PLMs would allow users to formulate queries to knowledge bases (KBs) in natural language, which makes them accessible to a wider user base (Heinzerling and Inui, 2021). Despite recent advances enabling smooth conversational interactions, e.g., with ChatGPT[1], factuality is still an open issue (Ray, 2023).

Many methods and datasets have been proposed to *probe* PLMs for factual knowledge. Probing involves a PLM and a dataset. The dataset contains truthful facts. These facts are used to estimate the amount of knowledge in PLMs. More specifically, the dataset contains inputs that identify the fact we are looking for, in order to extract it from the PLM (e.g., "Dante was born in [MASK]"), and ground

---

[1]https://openai.com/blog/chatgpt

truth answers that help evaluate if the retrieved answers are indeed correct (e.g., Florence). The data is often described in terms of relations (e.g., "place-of-birth") between subjects (e.g., "Dante") and objects (e.g., "Florence"). To produce prompts, a template is created for each relation (e.g., '[X] was born in [MASK]"), that is then filled with subject entities. The inputs can also have other forms such as questions (e.g., "Where was Dante born?").

In this work, we review recent work about factual knowledge probing. For the survey, we considered papers that cite the seminal work by Petroni et al. (2019) which first introduced the concept of PLMs as KBs.[2] We make the following contributions: (1) We provide a categorization of factual knowledge probing methods that is based on how inputs, PLMs and their outputs are adapted (see Figure 1 and Section 2); (2) We provide an overview of the datasets used for factual knowledge probing and categorize these under three classes based on their goal (Section 3); (3) We synthesize insights about knowledge retention and prompt optimization in PLMs (Section 4), analyze obstacles to adopting PLMs as knowledge bases (Section 5), and outline directions for future work (Section 7). We make our corpus of relevant papers publicly available.

## 2 Methods for Factual Probing

We categorize factual probing methods based on adaptations to i) input, ii) model, and iii) output. Categories are not mutually exclusive, i.e., one method could adapt input and model simultaneously. Figure 1 and Table 1 provide an overview of the probing methods. We only consider prompting methods that have been explicitly used for factual knowledge probing. For a general review of prompting methods, we refer to (Liu et al., 2023).

### 2.1 Probing Inputs

We distinguish between non-optimized or fixed inputs, and optimized inputs that are adapted in various ways to elicit more facts from PLMs.

### 2.1.1 Non-optimized Inputs

Extracting factual knowledge from PLMs depends on providing them with short inputs that indirectly describe the sought-after information. These methods can take various forms (cloze prompts (Taylor, 1953), questions, or entities). Non-optimized inputs represent the simplest case, where the probing

---

[2]For more details refer to Appendix A.1

inputs are not altered in any way.

**Cloze prompts** are widely used across several methods. Petroni et al. (2019) probe PLMs for factual knowledge by manually constructing cloze-style templates for several relations. Onoe et al. (2022) automatically construct cloze prompts from Wikipedia and Wikidata by masking out spans near entities of interest, in order to evaluate PLMs' knowledge about (unseen) entities. Abaho et al. (2022) construct cloze prompts from annotated PubMed abstracts to use PLMs as health outcome predictors. Chen et al. (2022) finetune PLMs using cloze prompts that consist of task descriptions alongside a few examples to elicit more facts.

**Questions** are the second input category. Several Question Answering datasets are used to finetune T5 models (Raffel et al., 2020), and evaluate the amount of knowledge implicitly present in their parameters in (Roberts et al., 2020). Multiple choice questions are used in (Hardalov et al., 2020) by providing PLMs with the questions followed by each option individually. The options are masked, and the final answer is selected based on the normalized log probabilities of the predicted tokens for each option. Kalo and Fichtel (2022) present a dataset based on Wikipedia, where inputs consist of several questions and answers, i.e., a few examples to implicitly indicate the task, and a similar question without an answer for evaluation.

**Entities** are used in methods that infer relational information or generate descriptions based on these entities. Some methods depend on a simple classifier or cosine similarity between the subject and object representations to determine the presence or absence of a relation. For example, to probe for geographical knowledge, Liétard et al. (2021) use fixed inputs that contain locations (e.g., countries or cities). These inputs are then used to extract representations for the respective locations from PLMs. Using these representations, the authors evaluate based on the ability of a simple classifier to solve certain tasks (e.g., predicting if two countries share border). Dufter et al. (2021) evaluate the amount of knowledge present in static word embeddings by matching a subject entity (the query) to an object entity from a pre-defined set of possible objects based on the cosine similarity between the representations of the subject and object entities. Shi et al. (2021) train generative PLMs to generate entities' descriptions while providing only the en-

tities as inputs, and compare them to ground truth descriptions.

### 2.1.2 Optimized Inputs

Probing inputs contribute substantially to the probing procedure. PLMs are sensitive to the inputs (Petroni et al., 2019; Jiang et al., 2020b; Elazar et al., 2021), and even syntactical variations or distractors, that do not alter the meaning, cause the PLM's predictions to change (Heinzerling and Inui, 2021; Longpre et al., 2021; Pandia and Ettinger, 2021; Podkorytov et al., 2021; Li et al., 2022a). Therefore, depending on the probing inputs, the estimate on factual knowledge we obtain may vary significantly. Optimized inputs represent variations of the inputs, where the inputs are changed to account for the sensitivity of the probed PLMs.

**Diversification and mining** methods aim to diversify and optimize prompts by mining Wikipedia or other resources, and selecting the best performing prompts or a combination of them. For example, Jiang et al. (2020b) propose a mining-based and a paraphrasing-based approach to create alternative prompts that outperform manual ones. The final prompts are selected based on their performance on a training set, and can also be combined in an ensemble. Bouraoui et al. (2020) mine for prompts that contain the entities of interest, and filter these based on the ability of the probed PLMs to predict the masked objects. After the filtering step, the remaining prompts are utilized to create a dataset that consists of positive inputs, i.e., containing true subject-object pairs, and negative inputs, which contain false pairs. This dataset is then used for the final evaluation.

**Direct optimization** methods aim to directly optimize existing prompts. This optimization happens either in a discrete space, to keep the prompts in natural language, or in a continuous space where the prompts do not have to correspond to specific tokens from the vocabulary. Optimization could also target only the masked token or the order of the examples in the prompt, in case a few examples are provided in the prompt to better indicate the task. Shin et al. (2020)'s AUTOPROMPT extends manually created prompts by prompts with a pre-defined number of trigger tokens, and employs gradient-based search to sequentially replace the trigger tokens with concrete tokens. These tokens are chosen to increase the probability of predicting the correct object. OPTIPROMPT (Zhong et al., 2021) is sim-

ilar to AUTOPROMPT, but allows for the trigger tokens to be replaced with vectors from a continuous embedding space. In a similar fashion, Qin and Eisner (2021) propose learning an ensemble of continuous prompts per relation. Additionally, they perturb the representations of the prompts in each layer in the probed PLMs using small learnable vectors. The intuition is to have activation patterns that are similar to the ones encountered during pre-training, which would make it easier to elicit knowledge from PLMs. Newman et al. (2022) utilize adapters (Houlsby et al., 2019) to map the embedding vectors to continuous prompts in order to make the probed PLMs less sensitive to different phrasings of the same prompts. Saeed and Papotti (2022) augment the masked tokens with a special type of embeddings, called Type Embeddings. These embeddings are derived from several entities that share the same type, and are shown to help tie the probed PLM's predictions to the expected type of the masked entity. PERO (Kumar and Talukdar, 2021) depends on querying PLMs with prompts containing few training examples (or shots), which demonstrate the task to the queried PLMs. Since PLMs are quite sensitive to the order and the quality of the provided training examples in the prompt, PERO leverages a genetic algorithm to find an optimized prompt and a separator token to concatenate the examples in the prompts. (Li et al., 2022c) exploit the symmetry of the task, and optimize prompts in a continuous space so that the probability of predicting both the subject and the object is maximized using the resulting prompts.

**Generation with PLM** methods re-write prompts with the help of a secondary PLM. Haviv et al. (2021) re-write manual prompts using another version of the probed model. The re-writing model is trained to produce prompts that help extract more knowledge from the probed one, which is kept unchanged. Zhang et al. (2022) leverage a generative PLM to produce optimized prompts.

## 2.2 Probed PLMs

PLMs are probed for knowledge using either their original pre-trained parameters (Petroni et al., 2019; Jiang et al., 2020b), or after adapting these parameters (Roberts et al., 2020; Meng et al., 2022b).

### 2.2.1 Vanilla PLMs

Methods in this category do not induce any changes to the probed PLMs, and depend on pre-training ob-

jectives to probe PLMs for factual knowledge. Using the pre-trained parameters is the most straightforward approach and is claimed to preserve the facts learned during pre-training (Elazar et al., 2021; Newman et al., 2022).

Most methods leverage the language modeling objectives from pre-training to probe for factual knowledge (Petroni et al., 2019; Jiang et al., 2020b; Shin et al., 2020; Haviv et al., 2021; Kumar and Talukdar, 2021; Zhong et al., 2021; Kalo and Fichtel, 2022; Newman et al., 2022; Onoe et al., 2022; Saeed and Papotti, 2022). Other methods rely on representations that come from the model's body, discarding task-specific parameters altogether (e.g., the Masked Language Modeling head in BERT-like models) (Liétard et al., 2021) or use representations of the subject and object entities in the case of static word embeddings (Dufter et al., 2021).

### 2.2.2 Adapted PLMs

Some works adapt the PLMs under evaluation to enable evaluation tasks, that do not correspond to any pre-training objective. The adaptation, however, is also coupled with risks such as train-test overlap (Lewis et al., 2021; Wang et al., 2021a).

**Supervised adaptation.** Most methods finetune the probed PLMs in a supervised manner to adapt them to the probing task. Roberts et al. (2020) finetune T5 models for closed-book question answering, where models have only questions as inputs, while leaving out any context or external knowledge sources that might contain the answer. Similarly, Wang et al. (2021a) finetune BART to output a related passage, and then the answer. Bouraoui et al. (2020) finetune BERT to classify prompts based on whether the relation between the subject and object entities truly holds or not. Fichtel et al. (2021) finetune a BERT model with its masked language modeling head to predict the masked tokens in the provided prompts. Abaho et al. (2022) propose an additional position-attention layer on top of transformer models, where the position of the masked token is kept constant, and the remaining tokens are given positions relative to the masked token. This approach is considered to put more focus on the masked tokens and its interaction with the remaining tokens in the prompt. Chen et al. (2022) leverage a task description that depends on the relation between the subject and object entity, alongside a few labeled examples to train the probed PLMs. At inference time, the PLMs are kept frozen

and are provided with unseen task descriptions and labeled examples to adapt to the task. Elazar et al. (2021) further train BERT with a consistency loss to increase its robustness to paraphrases that describe the same relation. Shi et al. (2021) finetune generative PLMs to generate entity descriptions depending only on their knowledge from pre-training. Qin and Eisner (2021) do not directly change any parameters in PLMs, but rather introduce additional trainable parameters in each layer that change the hidden representations of the prompts to help make them more suitable for knowledge extraction.

**Self-supervised adaptation.** Adaptations in a self-supervised manner can introduce changes to the model without explicitly finetuning the model to the probing task. For example, Meng et al. (2022b) propose to *re-wire* the probed PLM in a self-supervised manner. Their method depends on using data from the pre-training phase, splitting each sentence into a head part and a tail part, and using a contrastive learning objective to push the representations of the matching head and tail pairs (positives) closer to one another, and that of the non-matching pairs (negatives) to be further apart. The evaluation is based on the similarity between the representations of the prompt and a predefined set of entities that represent potential answers.

### 2.3 Outputs

Methods focusing on the outputs of PLMs address restricting the output space of PLMs, debiasing their outputs, and handling multi-token entities.

**Typed querying.** Kassner et al. (2021) propose to restrict the space of possible values for replacing the masked token (object) from the whole vocabulary to a specific set of tokens whose type matches the type of the ground truth object. For example, if the PLM is queried with the prompt: "The smallest country in the world is [MASK]", only entities of type country are considered to replace the [MASK] token. This method has two advantages: it reduces the number of objects under consideration and allows for a better comparison across PLMs with different vocabularies (Kassner et al., 2021).

**Debiasing.** Zhao et al. (2021) identify biases in the predictions of PLMs towards common and recent tokens, and propose a method that adapts the output probabilities by first estimating these biases using neutral examples and then correcting them. This debiasing method is shown to reduce

the variance across prompts and has a positive effect on fact retrieval. Malkin et al. (2022) propose a method to increase the effect of distant tokens on the predictions of PLMs. The method depends on combining two output distributions over the vocabulary. One distribution is based on the full-length input, whereas the other is based on a shortened version of the same input. Wang et al. (2023) identify the problem of object bias in optimized prompts and propose to make all potential objects equally probable when no subject is provided, and increasing the probability of the correct object, when the subject is available. Yoshikawa and Okazaki (2023) output predictions only above a sufficient confidence threshold. This results in a less biased evaluation, and reflects the ability of PLMs in excluding uncertain predictions. To address the problems of multiple valid answers and frequency bias, i.e., the co-occurence of some subject and object entities despite not being in a factual relation to one another, Dong et al. (2022) use two templates, one contains the correct relation while the other contains an erroneous relation between the two entities, and compare the probability for the correct object under both relations.

**Multi-token entities.** To handle multi-token entities, Jiang et al. (2020a) propose using a predefined number of masked tokens and filling these using different strategies: 1) independent from each other, 2) sequentially (left-to-right for English), 3) starting with the most confident predictions. (Kalinsky et al., 2023) leverage the masked token representation to generate multiple tokens using a small generative model.

## 3 Datasets for Factual Probing

We found a variety of datasets (44 in our corpus) that have been proposed or used for probing factual knowledge in PLMs: 18 datasets for probing general knowledge, 8 for domain-specific knowledge and 18 datasets that target other aspects, e.g, consistency of PLMs (cf. Table 2).

Datasets for **general knowledge** probing are used to quantify generic factual knowledge in PLMs with the most prominent being LAMA (Petroni et al., 2019). WIKI-UNI (Cao et al., 2021) is similar to LAMA, but with a uniform distribution of object entities. LAMA-UHN (Poerner et al., 2020) is a subset of LAMA without easy-to-guess examples. DLAMA (Keleg and Magdy, 2023) targets culturally diverse

facts. While 16 datasets are solely English, there are three multilingual datasets (mLAMA (Kassner et al., 2021), X-FACTR (Jiang et al., 2020a) and DLAMA (Keleg and Magdy, 2023)). IndicGLUE (Kakwani et al., 2020) contains 11 Indic languages. Most datasets consist of cloze prompts, while QA datasets (WebQuestions (Berant et al., 2013), TriviaQA (Joshi et al., 2017), NQ (Kwiatkowski et al., 2019)), PopQA and EntityQuestions (Mallen et al., 2023) are also used to quantify factual knowledge (Roberts et al., 2020). Wang et al. (2021a) adapt SQuAD (Rajpurkar et al., 2018) for closed-book question answering.

6 out of 8 datasets used for probing **domain-specific** knowledge target the biomedical domain (e.g., MedQA (Jin et al., 2021), BioLAMA (Sung et al., 2021) and MedLAMA (Meng et al., 2022b)). The multilingual dataset EX-AMS (Hardalov et al., 2020) focuses on scientific QA, whereas LEFT (Ciosici et al., 2021) contains questions from humanities and social sciences.

The community has constructed further datasets to investigate **other aspects** of using PLMs as knowledge bases. PARAREL (Elazar et al., 2021) and its multilingual counterpart mPARAREL (Fierro and Søgaard, 2022) target the sensitivity of PLMs to paraphrases. Negated/Misprimed LAMA (Kassner and Schütze, 2020) focuses on how negation/mispriming affects fact retrieval from PLMs, whereas Pandia and Ettinger (2021) target the effect of distractors. Updating knowledge in PLMs is considered by Jang et al. (2022a,b); Lee et al. (2022); Meng et al. (2022a); Hase et al. (2023); Hoelscher-Obermaier et al. (2023); Margatina et al. (2023). TEMPLAMA (Dhingra et al., 2022) is concerned with time-dependent facts retrieval, whereas SituatedQA (Zhang and Choi, 2021) considers both, temporal and geographical contexts. Heinzerling and Inui (2021) use a large dataset to evaluate the knowledge storing and retrieval capabilities of PLMs, and hence their use as KBs. Singhania et al. (2022) challenge the community to build a KB from PLMs, and provide a dataset to facilitate fact retrieval.

## 4 Insights about Knowledge Retention and Prompt Optimization

Two further aspects emerged from the surveyed papers: i) factors affecting knowledge retention, and ii) whether prompts should be optimized.

| Paper | Input | Opt. | Adapt. | Example | Tested PLMs | Eval. |
|---|---|---|---|---|---|---|
| Petroni et al. (2019) | C | | | Dante was born in [MASK]. | fairseq-fconv, ELMo, Transformer-XL, BERT | p@k |
| Bouraoui et al. (2020) | C | DM | ✓ | mining trigger prompts
[X] is the capital of [Y]. | BERT | F1 |
| Hardalov et al. (2020) | Q | | +✓[a] | <Q> → <A1,A2,A3,A4> | XLM-R | p@1 |
| Jiang et al. (2020a) | C | MT | | Barack Obama is a [MASK] [MASK] [MASK] by profession. | mBERT, XLM, XLM-R | p@1 |
| Jiang et al. (2020b) | C | DM | | prompt mining and paraphrasing
DirectX is developed by [MASK]. [MASK] released DirectX. DirectX is created by [MASK]. | BERT, ERNIE, KnowBert | p@1 |
| Roberts et al. (2020) | Q | | ✓ | Who lives in the imperial palace in Tokyo? | T5 | EM |
| Shin et al. (2020) | C | DO | | prompts optimization in discrete space
[X] is memory arcade branding by [MASK] | BERT, RoBERTa | p@1, p@10, MRR |
| Dufter et al. (2021) | E | | | sim(<capital entity>, <country entity>) | BERT, mBERT, fastText | p@1 |
| Elazar et al. (2021) | C | | ✓ | trains PLM with consistency loss
The capital of Italy is [MASK], Italy's capital, [MASK]. | BERT | p@1, cons, cacc |
| Fichtel et al. (2021) | C | | ✓ | Dante was born in [MASK]. | BERT | p@1 |
| Haviv et al. (2021) | C | LM | | re-writing with PLM
will & grace is originally aired on [MASK]. | BERT | p@1 |
| Kassner et al. (2021) | C | TY | | Berlin is the capital of [MASK]$_{country}$ | BERT, mBERT | p@1 |
| Kumar and Talukdar (2021) | C | DO | | examples reordering
ex1, ex2, ex3, Rome is located in [MASK]. | BERT | p@1 |
| Liétard et al. (2021) | E | | | He lives in <location entity>. | BERT, RoBERTa, GPT-2 | PER |
| Qin and Eisner (2021) | C | DO | ✓ | prompts optimization in continuous space, perturbations of representations in all layers
[X] [V1] ... [V5] [MASK] [V6] | BERT, RoBERTa | p@1, p@10, MRR |
| Shi et al. (2021) | E | | ✓ | generating descriptions for entities
[Carl Menger] was an Austrian... | BART, T5 | R-L |
| Wang et al. (2021a) | Q | | ✓ | <Q> → <answer related passage> <A> | BART | EM, F1, HE |
| Zhao et al. (2021) | C | DEB | | estimates and corrects biases
NA was born in [MASK]. | GPT-3 | p@1 |
| Zhong et al. (2021) | C | DO | | prompts optimization in continuous space
[X] [V1] ... [V5] [MASK] | BERT | p@1 |
| Abaho et al. (2022) | C | | ✓ | Two CMZ patients and one morphine patient showed complete [MASK]. | BERT, BioBERT, Biomed_RoBERTa, SciBERT, UmlsBERT | EM, PM |
| Chen et al. (2022) | C | | ✓ | <task description> <example>* Dante was born in [MASK]. | BERT, DeBERTa | p@1 |
| Dong et al. (2022) | C | DEB | | uses probabilities for correct/incorrect relations
P(Hawaii \| Obama was born in) / P(Hawaii \| Obama worked) | T5 | False rate |
| Kalo and Fichtel (2022) | Q | | | <Q&A>*, What languages does Confuzius speak? | GPT-J, GPT-2, OPT | F1 |
| Li et al. (2022c) | C | DO | | optimized prompts to predict subject and object
([MASK]) [V1] ... [V5] ([MASK]) | BERT, RoBERTa | p@k, MRR |
| Malkin et al. (2022) | C | DEB | | combines two output distributions
Dante was born in [MASK], was born in [MASK] | GPT-2, GPT-3 | p@1 |
| Meng et al. (2022b) | C | | ✓* | sim(Elvitegravir may prevent [MASK], entity) | BERT, BlueBERT, BioBERT, T5, BART, PubMedBERT, SciFive | p@1 |
| Newman et al. (2022) | C | DO | | adapter mapping prompts to continuous prompts after embedding layer
[V1] ... [V5] from [MASK] is Canada's capital | BERT | p@1, cons |
| Onoe et al. (2022) | C | | | [mRNA vaccines] do not affect [MASK]. | T5, BART, GPT-Neo | pplx |
| Saeed and Papotti (2022) | C | DO | | masked tokens with type embeddings
The wife of Obama is ([MASK] + [TE]). | BERT | p@1, p@k |
| Zhang et al. (2022) | C | LM | | generating prompts by PLM (BART)
Marco Benevento and not violin yeah much like trafficking UNESCO partly [MASK]. | BERT | p@1 |
| Kalinsky et al. (2023) | C | MT | | uses the masked token repr. to generate multi-token predictions
I love [MASK] city. | BERT | p@1 |
| Wang et al. (2023) | C | DEB | | reduces object bias
The native language of [X] is [MASK]. | BERT, RoBERTa | p@1, MRR, entropy |
| Yoshikawa and Okazaki (2023) | C | DEB | | outputs prediction by sufficient confidence
[X] was born in [MASK]. | BERT, RoBERTa | p@1, RC-AUC |

Table 1: Overview of probing methods. Input type: cloze prompts C, questions Q, entities E. Prompt optimization: diversification and mining DM, direct optimization DO, or generation with PLMs LM. Other methods: debiasing DEB, mutli-token entities MT, or typed querying TY. PLM adaptation: supervised (✓), or self-supervised (✓*). Evaluation: consistency (cons), consistent-accuracy (cacc), exact match (EM), human evaluation (HE), mean reciprocal rank (MRR), partial match (PM), perplexity (pplx), probe error reduction (PER), ROUGE-L (R-L), and AUC of the risk-coverage curve (RC-AUC).

---

[a]adapted and non-adapted PLM

| | Dataset | Cat. | Lang. | Example | #Inst. | Access |
|---|---|---|---|---|---|---|
| **GENERAL KNOWLEDGE** | LAMA (Petroni et al., 2019) | GK C | en | Dante was born in [MASK] | 40k | + |
| | Google Analogy(semantic) (Bouraoui et al., 2020) | GK CLS | en | It is located in [X], the capital of [Y] | 9k | + |
| | WebQuestions (Roberts et al., 2020) | GK Q | en | What degrees did Obama get? | 6k | + |
| | BATS (ency.) (Bouraoui et al., 2020) | GK CLS | en | [X] is the capital of [Y] | 0.5k | + |
| | TriviaQA (Roberts et al., 2020) | GK Q | en | Who won the Nobel Peace Prize in 2009? | 96k | + |
| | NQ (Roberts et al., 2020) | GK Q | en | Who lives in the imperial palace in Tokyo? | 322k | + |
| | IndicGLUE (Kakwani et al., 2020) | GK C | indic[a] | Shambhupara <MASK> is an important village in Amreli Tehsil, Gujarat State. | 239k | + |
| | X-FACTR (Jiang et al., 2020a) | GK C | multi | The mother tongue of Obama is [MASK] | 398k | + |
| | LAMA-UHN (Poerner et al., 2020) | GK C | en | USA maintains diplomatic relations with [MASK] | 32k | o |
| | LPAQA (Jiang et al., 2020b) | GK C | en | DirectX is developed/created by [MASK] | 3k | + |
| | mLAMA (Kassner et al., 2021) | GK C | multi | Paris is the capital of [MASK] | 855k | + |
| | DESCGEN (Shi et al., 2021) | GK NLG | en | [Carl Menger] was an Austrian economist... | 37k | + |
| | WIKI-UNI (Cao et al., 2021) | GK C | en | Turing was born in [MASK]. | 70k | + |
| | SQuAD (Wang et al., 2021a) | GK Q | en | <Q> → <answer related passage> <A> | 92k | + |
| | KAMEL (Kalo and Fichtel, 2022) | GK Q | en | <Q&A>*, What languages does Confuzius speak? | 47k | + |
| | DLAMA (Keleg and Magdy, 2023) | GK C | multi | Egypt is located in [MASK] | 78k | + |
| | PopQA (Mallen et al., 2023) | GK Q | en | What is the capital of Louisiana? | 14K | + |
| | EntityQuestions (Mallen et al., 2023) | GK Q | en | Who is the author of The Target? | 177k | + |
| **DOMAIN-SPECIFIC** | EXAMS (Hardalov et al., 2020) | DK Q | multi | <Q> <A1,A2,A3,A4> → <A$_i$> | 24k | + |
| | MedQA (Jin et al., 2021) | DK Q | en,zh[b] | <Case> <Q> <A1,A2,A3,A4> → <A$_i$> | 61k | + |
| | DisKnE (Alghanmi et al., 2021) | DK CLS | en | The patient has high BP <SEP> Hypertension | 7k | o |
| | (Yuan et al., 2021) | DK C | en | apraclonidine may prevent [MASK] | 144k | o |
| | LEFT (Ciosici et al., 2021) | DK CLS | en | <statement> → <True/False> | 1k | o |
| | BioLAMA (Sung et al., 2021) | DK C | en | Hepatitis has symptoms such as [MASK] | 49k | + |
| | EBM-NLP (Abaho et al., 2022) | DK C | en | ...patient showed complete [MASK] | 3k | - |
| | MedLAMA (Meng et al., 2022b) | DK C | en | Elvitegravir may prevent [MASK] | 19k | + |
| **OTHER** | Negated LAMA (Kassner and Schütze, 2020) | CO C | en | The capital of Italy is not [MASK] | 10k | + |
| | Misprimed LAMA (Kassner and Schütze, 2020) | CO C | en | Dinosaurs? Munich is located [MASK] | 11k | + |
| | ParaRel (Elazar et al., 2021) | CO C | en | Turing was born in/is native to [MASK] | n.a.[c] | + |
| | (Pandia and Ettinger, 2021) | CO C | en | Sebastian lives in France. The capital of Sebastian's country is [MASK]. | 40k | + |
| | SituatedQA (context/answer) (Zhang and Choi, 2021) | CK Q | en | Who made the most 3 point shots in the NBA? | 18k | + |
| | (Heinzerling and Inui, 2021) | KB C | en | Turing was born in [MASK] | 15M | + |
| | (Podkorytov et al., 2021) | MC C | en | Tomatoes are a [MASK]. | 0.1k | - |
| | mParaRel (Fierro and Søgaard, 2022) | CO C | multi | Turing is from/was born in [MASK] | n.a.[d] | + |
| | TEMPLAMA (Dhingra et al., 2022) | CK C | en | [2012] Cristiano Ronaldo plays for [MASK] | 50k | + |
| | (Singhania et al., 2022) | KB C | en | France shares a land border with [MASK] | 2k | + |
| | (Jang et al., 2022b) | KU C | en | [MASK] is the prime minister of England | 30k | + |
| | TemporalWiki (Jang et al., 2022a) | KU PPL | en | On 1 December, the Omicron variant... | ⓓ | o |
| | zsRE (Lee et al., 2022) | KU Q | en | Who is the most paid player in EPL? | 168k | + |
| | CounterFact (Meng et al., 2022a) | KU C | en | Turing's mother tongue is <old,new> | 22k | + |
| | ECBD (Onoe et al., 2022) | UE C | en | [mRNA vaccines] do not affect [MASK]. | 35k | + |
| | (Hase et al., 2023) | KU C | en | Mary Lowe Good has relation 'winner of' to [MASK] | 170k | + |
| | CounterFact+ (Hoelscher-Obermaier et al., 2023) | KU C | en | The mother tongue of Danielle Darrieux is English. The native language of Montesquieu is [MASK] | ⓓ | o |
| | DynamicTempLAMA (Margatina et al., 2023) | KU C | en | The surname of the Prime Minister of the UK is [MASK] | ⓓ | + |

Table 2: Datasets for factual knowledge probing. Probed knowledge: general knowledge GK, domain-specific knowledge DK, context-dependent knowledge CK, PLMs sensitivity to paraphrases, negation or mispriming CO, related to PLMs as KBs KB , knowledge updating KU, misconceptions MC and unseen entities UE. NLP task: cloze prompts C, question answering Q, classification CLS, natural language generation NLG, and perplexity PPL. Showing languages, example, and number of instances in the dataset (rounded). Data access: accessible without effort (+), accessible with some effort (o), not accessible (-). ⓓ refers to dynamic datasets, whose number of instances is changeable over time. We only include references to papers, in which the datasets are used for factual knowledge probing. References to papers introducing the datasets are added in Table 4 in the appendix.

---

[a]11 different languages

[b]including zh-simplified

[c]number of relations 328, prompts per relation 38

[d]number of relations 343, prompts per relation 37.13 (avg. over languages)

## 4.1 Factors Affecting Knowledge Retention

PLMs are diverse with respect to their architectures, pre-training objectives and their pre-training data. A compelling question is: how do all these factors affect knowledge retention in PLMs?

Large language models are known to perform generally better and hold more knowledge (Brown et al., 2020; Roberts et al., 2020). However, the model's architecture and pre-training objectives are more decisive for knowledge retention than its size (Li et al., 2022a). For example, pre-training with the Salient Span Masking objective (Guu et al., 2020) helps PLMs to absorb more facts (Roberts et al., 2020; Cole et al., 2023). Similarly, Xiong et al. (2020) demonstrate that training the model to predict if the original entities in the text have been replaced with other entities is beneficial for fact retrieval. More generally, Ye et al. (2021) conclude that a masking strategy matching the downstream task, positively affects the performance on that task.

A larger pre-training corpus with an encoder-only model (Liu et al., 2020) leads to higher knowledge retention (Zhang et al., 2021), but with an encoder-decoder model (Lewis et al., 2020), a larger corpus negatively affects knowledge retention Wang et al. (2021a). Recency (Chiang et al., 2020) and frequency (Kandpal et al., 2023), i.e., *when* and *how often* the data is observed at training, are also essential for knowledge retention.

Larger models and more pre-training data can improve knowledge retention if combined with the right choices for architecture and pre-training objective(s). However, scaling might not be sufficient (Kandpal et al., 2023). Even though many works propose new architectures and pre-training objectives to increase factual knowledge retention in PLMs and their robustness to prompts (Févry et al., 2020; Hosseini et al., 2021; Sadeq et al., 2022; Whitehouse et al., 2022; Min et al., 2023; Zhong et al., 2023), this is a promising future work direction, as there is more room for improvement.

## 4.2 Should Prompts be Optimized?

Prompt Optimizing leads to better probing performance (Jiang et al., 2020b; Shin et al., 2020; Kumar and Talukdar, 2021; Newman et al., 2022; Zhang et al., 2022) . However, it remains unclear whether this improvement is due to optimized prompts leaking new knowledge into the probed PLMs.

Optimized prompts can be mere paraphrases of manually created prompts (Bouraoui et al., 2020; Jiang et al., 2020b). These paraphrases might be better fact retrievers because of their similarity to the pre-training corpus (Cao et al., 2022). Other prompt optimization methods find better prompts in discrete or continuous spaces (Shin et al., 2020; Zhong et al., 2021). These prompts are largely uninterpretable, and can even retrieve facts from randomly initialized PLMs (Zhong et al., 2021; Ishibashi et al., 2023).

Performance improvements for optimized prompts can be attributed either to prompts becoming more similar to the pre-training data or overfitting the facts distribution. Evaluation should take the pre-training corpora and the facts distribution in the probing dataset into account (Cao et al., 2021, 2022). Future work should consider adapting prompt optimization methods to produce more interpretable prompts. This would keep the performance gains, and increase the trustworthiness of optimized prompts.

## 5 Obstacles to Adopting PLMs as KBs

**Consistency.** A challenge to relying on PLMs as knowledge bases is their sensitivity to the input queries (Fierro and Søgaard, 2022). PLMs rely on shallow surface features and lexical correlations (Kassner and Schütze, 2020; Misra et al., 2020; Poerner et al., 2020; Rogers et al., 2020; Li et al., 2022b), which explains their high sensitivity to the way queries are formulated. Current solutions (Elazar et al., 2021; Newman et al., 2022) train PLMs to be robust to variations in inputs, but further improvements are needed to make PLMs reliable knowledge bases. PLMs are known to be highly sensitive to prompts, especially in languages other than English (Fierro and Søgaard, 2022), where less resources are available. Making PLMs more robust to prompts in non-English languages is a promising future work direction.

**Interpretability.** Identifying where facts are stored and how they are retrieved is essential to adopt PLMs as trustworthy knowledge sources. Several approaches locate knowledge in PLMs (Wallat et al., 2020; Podkorytov et al., 2021; Alkhaldi et al., 2022; Dai et al., 2022; Meng et al., 2022a), with different conclusions depending on the architecture (e.g., knowledge is located in the middle layers of GPT-like models (Meng et al., 2022a), or in the upper layers in BERT-like models (Dai et al., 2022)). Another line of work focuses on the data aspect, showing the dependence

of PLMs on word co-occurrences and positionally close words (Li et al., 2022b), or tracing back predictions to training data (Akyurek et al., 2022; Park et al., 2023). Knowing how PLMs retrieve facts remains challenging, but necessary to make PLMs transparent fact retrievers. The introduction of a fact tracing benchmark (Akyurek et al., 2022) opens the door for works in this direction.

**Updating Knowledge.** PLMs come with a fixed set of pre-trained parameters that encode knowledge about the world. As time passes, this knowledge becomes partially outdated. Hence, editing existing knowledge in PLMs and augmenting them with new knowledge is crucial for their use as knowledge bases (Zini and Awad, 2022).

One line of research locates the modules responsible for factual predictions and modifies these to update the corresponding facts (Dai et al., 2022; De Cao et al., 2021; Meng et al., 2022a). Other lines of research keep the original PLM unchanged, but augment it with additional parameters to induce the desired changes (Wang et al., 2021b; Lee et al., 2022), or encode facts with time stamps in PLMs to make them "time-aware" (Dhingra et al., 2022).

When updating facts in PLMs, it is crucial that only the targeted facts are affected and that these facts are retrievable using different paraphrases (De Cao et al., 2021; Hase et al., 2023). However, current methods for facts editing (Meng et al., 2022a, 2023) still do not fulfill these requirements (Hoelscher-Obermaier et al., 2023). Methods that introduce additional parameters should be made more scalable (Jang et al., 2022b).

## 6 Related Work

AlKhamissi et al. (2022) elaborate requirements for PLMs as knowledge bases and review recent literature w.r.t. those requirements. These requirements are widely known (e.g., consistency (Petroni et al., 2019) and updating knowledge (De Cao et al., 2021)). Our analysis leads to similar general observations (cf. Section 5), and additionally reviews more recent solutions to these obstacles. Cao et al. (2023) cover probing PLMs as part of the knowledge cycle in PLMs, but do not address factual knowledge probing at the same level of detail as we do. Liu et al. (2023) survey prompting methods in detail. However, they cover only a part of factual knowledge probing methods. Safavi and Koutra (2021) survey how PLMs acquire relational knowledge, organizing knowledge representations

strategies in PLMs based on different levels of KBs supervision. We provide a novel categorization scheme and conduct a systematic analysis of methods for factual knowledge probing that goes beyond all existing surveys. We additionally provide a categorization of factual probing datasets. Furthermore, we discuss recent findings on knowledge retention, the use of optimized prompts, and challenges with corresponding recent solutions to adopting PLMs as KBs, shedding light on several future work directions. In contrast to other work, we employed a systematic approach to curate and analyze relevant literature to a comprehensive and unbiased representation of existing work.

## 7 Discussion and Future Work

Factual probing methods are developed to extract as many facts as possible from the new smart pools of knowledge, namely PLMs. This gives us an estimate about how much PLMs have learned from pre-training, and help us to assess their suitability for use cases such as PLMs-as-KBs. Improving probing methods should go hand-in-hand with advances in PLMs themselves, to help us better assess and make use of PLMs. Our analysis (cf. Section 2) shows that current probing methods focus mostly on one the the three dimensions we use in our categorization (inputs, PLMs, outputs). Introducing adaptations across two or more of these dimensions (e.g., optimizing inputs while also debiasing outputs) might lead to further improvements with respect to factual knowledge retrieval.

Besides improving probing methods, it is also essential to pay attention to the benchmark datasets. Some probing datasets are shown to be biased towards certain entities (Cao et al., 2021). Constructing unbiased probing datasets is crucial to have unbiased estimates of factual knowledge in PLMs. At the same time, developing comprehensive datasets which correspond to the capacity of the recently published large PLMs, e.g., (OpenAI, 2023; Penedo et al., 2023; Touvron et al., 2023), is an important future work direction.

We also believe that it is necessary for current evaluation schemes to not be limited to counting how often PLMs answer correctly. Instead, we call for a comprehensive evaluation that includes further important factors such as the number and frequency of the answers in the pre-training corpus, creation period of the pre-training corpus, model size, and the number of training epochs.

## 8 Limitations

For our corpus construction we relied on all the publications that cited (Petroni et al., 2019). Although this represents the first work that sparked the community's interest in the factual knowledge present in PLMs and their use as KBs, there might be parallel works or works that go into the same direction but do not directly cite Petroni et al. (2019)'s work, which are not included in our corpus. Additionally, we relied on the venue information provided by Semantic Scholar's API to filter out irrelevant publications. These information are not always accurate and might have affected our initial corpus.

In this work, we focused on works that revolve around factual knowledge, and excluded works that focus on other types of knowledge (e.g., linguistic knowledge and commonsense knowledge). However, there are methods that are used for other types of knowledge that could also be applied to factual knowledge and vice versa. We consciously excluded works that focused on other types of knowledge, but this does not mean that such methods are not applicable to factual knowledge probing.

## Acknowledgements

We thank Jan Trienes, and the three anonymous reviewers for their insightful comments on this work.

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

## A Corpus Creation & Annotation Methodology

### A.1 Paper Selection

We created our corpus of papers about factual knowledge probing by querying the Semantic Scholar API[3] on 31. August, 2023 at 2:51 pm for all works citing (Petroni et al., 2019), resulting in 1,416 papers. According to our knowledge Petroni et al. (2019)'s work was the first to quantify factual knowledge in PLMs, and envisioned using PLMs as KBs. We then separated the papers based on their venue information: the venue papers (1,006 instances), the arXiv papers (375 instances) and the no-venue papers (35 instances) with missing venue information returned by the Semantic Scholar API. We did a web search for the no-venue papers and manually assigned them to their respective group, yielding a total of 1,008 venue papers. We then proceeded to annotate in two steps, after which our final corpus of papers were reduced to 94 highly relevant papers.

### A.2 First Annotation Step

Three authors single annotated the peer-reviewed papers as either relevant or not-relevant based on our inclusion and exclusion criteria, discussing uncertainties with all annotators. When in doubt, the paper was marked as relevant for a high recall.

A paper was marked as relevant based on its title, abstract or body, if it contained one of the following:

- Knowledge probing methods, metrics or datasets for quantifying the relational or factual knowledge stored in a PLM.
- Methods shown to increase the amount of relational or factual knowledge stored in a PLM.
- Methods to update, localize or increase the consistency of factual knowledge in PLMs.

We further explicitly included short and long papers, workshop papers, posters, shared tasks, competitions and challenge papers with at least 4 pages and explicitly excluded master theses, PhD dissertations, workshop proposals, workshop reports and

| 1 | Which probing methods are used? |
| 2 | Which probing methods are proposed? |
| 3 | Which PLMs are probed? |
| 4 | Are the probed PLMs fine-tuned? |
| 5 | Which knowledge probing datasets are used? |
| 6 | What are the sources for the datasets? |
| 7 | Does the paper present any methods or analysis with respect to factual knowledge probing (e.g., updability, interpretability, consistency)? |
| 8 | Does the paper propose a method to inject knowledge into a PLM? |
| 9 | Is the paper a survey? |

Table 3: The information extracted for each paper.

other non-peer-reviewed publications. After deduplication, the first annotation step yielded a total of 173 relevant papers.

### A.3 Second Annotation Step

We revisited all relevant papers and annotated them based on the questions listed in table 3. We only included publications that perform intrinsic evaluations (Kalyan et al., 2021) that directly target factual knowledge, and we exclude extrinsic evaluations on knowledge-intensive tasks. Additionally, we excluded papers that did not train on free-text corpora. For each probing method, we store its name, description and an example. For datasets, we store name, domain (of knowledge), original source (from which the dataset was compiled), number of instances and language of the dataset. We excluded 88 of 173 papers for being irrelevant upon second inspection and added 9 papers, which were not in our initial corpus of research papers. These papers introduced relevant datasets. Our final corpus of surveyed papers thus counts 94 highly relevant papers.

---

[3]https://api.semanticscholar.org/graph/v1/paper/d0086b86103a620a86bc918746df0aa642e2a8a3/citations?fields=intents,url,title,abstract,venue,year,referenceCount,citationCount,influentialCitationCount,fieldsOfStudy,publicationDate&limit=1000

| Dataset | Cat. | Lang. | Example | #Inst. | Access |
|---------|------|-------|---------|--------|--------|
| **GENERAL KNOWLEDGE** | | | | | |
| LAMA (Petroni et al., 2019) | GK C | en | Dante was born in [MASK] | 40k | + |
| (Bouraoui et al., 2020) Google Analogy (semantic) (Mikolov et al., 2013) | GK CLS | en | It is located in [X], the capital of [Y] | 9k | + |
| (Roberts et al., 2020) WebQuestions (Berant et al., 2013) | GK Q | en | What degrees did Obama get? | 6k | + |
| (Bouraoui et al., 2020) BATS (encyclopedic) (Gladkova et al., 2016) | GK CLS | en | [X] is the capital of [Y] | 0.5k | + |
| (Roberts et al., 2020) TriviaQA (Joshi et al., 2017) | GK Q | en | Who won the Nobel Peace Prize in 2009? | 96k | + |
| (Roberts et al., 2020) NQ (Kwiatkowski et al., 2019) | GK Q | en | Who lives in the imperial palace in Tokyo? | 322k | + |
| IndicGLUE (Kakwani et al., 2020) | GK C | indic[a] | Shambhupara <MASK> is an important village in Amreli Tehsil, Gujarat State. | 239k | + |
| X-FACTR (Jiang et al., 2020a) | GK C | multi | The mother tongue of Obama is [MASK] | 398k | + |
| LAMA-UHN (Poerner et al., 2020) | GK C | en | USA maintains diplomatic relations with [MASK] | 32k | o |
| LPAQA (Jiang et al., 2020b) | GK C | en | DirectX is developed/created by [MASK] | 3k | + |
| mLAMA (Kassner et al., 2021) | GK C | multi | Paris is the capital of [MASK] | 855k | + |
| DESCGEN (Shi et al., 2021) | GK NLG | en | [Carl Menger] was an Austrian economist... | 37k | + |
| WIKI-UNI (Cao et al., 2021) | GK C | en | Turing was born in [MASK]. | 70k | + |
| (Wang et al., 2021a) (Rajpurkar et al., 2018) | GK Q | en | <Q> → <answer related passage> <A> | 92k | + |
| KAMEL (Kalo and Fichtel, 2022) | GK Q | en | <Q&A>*, What languages does Confuzius speak? | 47k | + |
| DLAMA (Keleg and Magdy, 2023) | GK C | multi | Egypt is located in [MASK] | 78k | + |
| PopQA (Mallen et al., 2023) | GK Q | en | What is the capital of Louisiana? | 14K | + |
| (Mallen et al., 2023) EntityQuestions (Sciavolino et al., 2021) | GK Q | en | Who is the author of The Target? | 177k | + |
| **DOMAIN-SPECIFIC** | | | | | |
| EXAMS (Hardalov et al., 2020) | DK Q | multi | <Q> <A1,A2,A3,A4> → <A_i> | 24k | + |
| MedQA (Jin et al., 2021) | DK Q | en,zh[b] | <Case> <Q> <A1,A2,A3,A4> → <A_i> | 61k | + |
| DisKnE (Alghanmi et al., 2021) | DK CLS | en | The patient has high BP <SEP> Hypertension | 7k | o |
| (Yuan et al., 2021) | DK C | en | apraclonidine may prevent [MASK] | 144k | o |
| LEFT (Ciosici et al., 2021) | DK CLS | en | <statement> → <True/False> | 1k | o |
| BioLAMA (Sung et al., 2021) | DK C | en | Hepatitis has symptoms such as [MASK] | 49k | + |
| (Abaho et al., 2022) EBM-NLP (Nye et al., 2018) | DK C | en | ...patient showed complete [MASK] | 3k | - |
| MedLAMA (Meng et al., 2022b) | DK C | en | Elvitegravir may prevent [MASK] | 19k | + |
| **OTHER** | | | | | |
| Negated LAMA (Kassner and Schütze, 2020) | CO C | en | The capital of Italy is not [MASK] | 10k | + |
| Misprimed LAMA (Kassner and Schütze, 2020) | CO C | en | Dinosaurs? Munich is located [MASK] | 11k | + |
| ParaRel (Elazar et al., 2021) | CO C | en | Turing is from/was born in [MASK] | n.a.[c] | + |
| (Pandia and Ettinger, 2021) | CO C | en | Sebastian lives in France. The capital of Sebastian's country is [MASK]. | 40k | + |
| SituatedQA (context/answer) (Zhang and Choi, 2021) | CK Q | en | Who made the most 3 point shots in the NBA? | 18k | + |
| (Heinzerling and Inui, 2021) | KB C | en | Turing was born in [MASK] | 15M | + |
| (Podkorytov et al., 2021) | MC C | en | Tomatoes are a [MASK]. | 0.1k | - |
| mParaRel (Fierro and Søgaard, 2022) | CO C | multi | Turing is from/was born in [MASK] | n.a.[d] | + |
| TEMPLAMA (Dhingra et al., 2022) | CK C | en | [2012] Cristiano Ronaldo plays for [MASK] | 50k | + |
| (Singhania et al., 2022) | KB C | en | France shares a land border with [MASK] | 2k | + |
| (Jang et al., 2022b) | KU C | en | [MASK] is the prime minister of England | 30k | + |
| TemporalWiki (Jang et al., 2022a) | KU PPL | en | On 1 December, the Omicron variant... | ⊕ | o |
| (Lee et al., 2022) zsRE (Levy et al., 2017) | KU Q | en | Who is the most paid player in EPL? | 168k | + |
| CounterFact (Meng et al., 2022a) | KU C | en | Turing's mother tongue is <old,new> | 22k | + |
| ECBD (Onoe et al., 2022) | UE C | en | [mRNA vaccines] do not affect [MASK]. | 35k | + |
| (Hase et al., 2023) | KU C | en | Mary Lowe Good has relation 'winner of' to [MASK] | 170k | + |
| CounterFact+ (Hoelscher-Obermaier et al., 2023) | KU C | en | The mother tongue of Danielle Darrieux is English. The native language of Montesquieu is [MASK] | ⊕ | o |
| DynamicTempLAMA (Margatina et al., 2023) | KU C | en | The surname of the Prime Minister of the UK is [MASK] | ⊕ | + |

Table 4: Datasets for factual knowledge probing. Probed knowledge: general knowledge GK, domain-specific knowledge DK, context-dependent knowledge CK, PLMs sensitivity to paraphrases, negation or mispriming CO, related to PLMs as KBs KB, knowledge updating KU, misconceptions MC and unseen entities UE. NLP task: cloze prompts C, question answering Q, classification CLS, natural language generation NLG, and perplexity PPL. ⊕ refers to dynamic datasets, whose number of instances is changeable over time.

[a] 11 different languages
[b] including zh-simplified
[c] number of relations 328, prompts per relation 38
[d] number of relations 343, prompts per relation 37.13 (avg. over languages)