# OpenReview forum: "Give Me the Facts! A Survey on Factual Knowledge Probing in Pre-trained Language Models"
_EMNLP/2023/Conference — EMNLP 2023 Findings_

### Official Review · Reviewer_ASH9 · 2023-08-03

**Soundness:** 4

**Excitement:**

3: Ambivalent: It has merits (e.g., it reports state-of-the-art results, the idea is nice), but there are key weaknesses (e.g., it describes incremental work), and it can significantly benefit from another round of revision. However, I won't object to accepting it if my co-reviewers champion it.

**Paper Topic And Main Contributions:**

This paper survey methods and datasets that are used to probe PLMs for factual knowledge, which is the first to address the topic of factual knowledge probing in PLMs.

**Reasons To Accept:**

The authors propose a categorization scheme for factual probing methods and provide an overview of the datasets used for factual probing, then synthesize insights about knowledge retention and prompt optimization in PLMs, analyze obstacles to adopting PLMs as knowledge bases and outline directions.

**Reasons To Reject:**

Some representative works of this year are missing in this survey, and there are too many references to existing related surveys.

**Reproducibility:**

4: Could mostly reproduce the results, but there may be some variation because of sample variance or minor variations in their interpretation of the protocol or method.

**Reviewer Confidence:**

1: Not my area, or paper was hard for me to understand. My evaluation is just an educated guess.

---

> ### Author Rebuttal · Authors · 2023-08-29
>
> We thank Reviewer ASH9 for their feedback acknowledging that our survey is the first to address the topic of factual knowledge probing in PLMs. We address the reviewer's concerns (missing representative work, and existing surveys) below.
>
> In the following we address the reviewer’s concerns.
>
> ### The recency of the works covered in our survey:
> > **Some representative works of this year are missing in this survey,** and there are too many references to existing related surveys.
>
> The reviewer mentioned that our survey does not cover representative works from 2023. Indeed, we did not cover recent works from 2023. We will integrate novel work in the final version of the paper. To assess whether novel papers would impact the main message or structure of our paper, we systematically analyzed the papers of this year's passed main conferences EACL 2023 and ACL 2023. We found 9 relevant papers which we can easily integrate in our current structure. Our findings were the following:
> -  3 probing methods that target the outputs of PLMs. These fall in the **Outputs** category in our taxonomy of probing methods.
> - 3 datasets that target different aspects (culturally diverse facts, temporal drifts, editing facts in PLMs). These can also be added to the **General Knowledge** and **Other** datasets categories we have.
> - 2 papers that propose pre-training objectives to increase knowledge retention. These can be integrated in Section 4.1 **Factors Affecting Knowledge Retention**.
> - 1 paper that targets evaluating editing facts in PLMs. This fits into Section 5 under **Updating Knowledge**.
>
> ### How our work differs from other surveys:
> > Some representative works of this year are missing in this survey, **and there are too many references to existing related surveys.**
>
> The reviewer also remarked that “there are too many references to existing related surveys.” Since we only cite two related surveys in our work (see Introduction L82:85), we believe the reviewer was alluding to other related surveys that we did not cite. Here, we clarify how our survey differs from other surveys, extending the corresponding part in the paper pdf (L82:86 in Introduction) (for more details please check our response to Reviewer avt8):
>
> **Relation to the Surveys (paper version):**
> AlKhamissi et al. [2] elaborate requirements for PLMs as knowledge bases and review recent literature w.r.t. those requirements. These requirements are widely known (e.g., consistency [6] and updating knowledge [7]). Our analysis leads to similar general observations (cf. Section 5), and additionally provides more recent solutions to these obstacles (cf. Section 5 and  “Details on Relation to [2]”). Cao et al. [3] cover probing PLMs as part of the knowledge cycle in PLMs, but do not address factual knowledge probing at the same level of detail as we do. Liu et al. [4] survey prompting methods in detail. However, this covers only a part of factual knowledge probing methods. Safavi and Koutra [5] survey how PLMs acquire relational knowledge, organizing knowledge representations strategies in PLMs based on different levels of KBs supervision. In contrast to all existing surveys, we conduct a systematic analysis of methods for factual knowledge probing, providing a categorization scheme (that is admittedly novel and well-organized). We additionally provide a categorization of factual probing datasets. Furthermore, we discuss recent findings on knowledge retention, the use of optimized prompts, and challenges with corresponding recent solutions to adopting PLMs as KBs, shedding light on several future work directions. In contrast to other works [2,3,4,5], we employed a systematic approach to curate and analyze relevant literature. This ensured a comprehensive and unbiased representation of existing works.
>
>
> [1] Language Models As or For Knowledge Bases - Workshop on Deep Learning for Knowledge Graphs 2022  nonarxiv version https://ceur-ws.org/Vol-3034/paper2.pdf
>
> [2] A Review on Language Models as Knowledge Bases, arXiv version April 2022 - according to the first authors web-page ((https://bkhmsi.github.io)  this is under review at JAIR
>
> [3] The Life Cycle of Knowledge in Big Language Models: A Survey, arXiv version March 2023 - this might be under submission or accepted and not yet published, the first author's homepage (https://c-box.github.io) mentions "Machine Intelligence Research" for this paper
>
> [4] (reference [Liu et al., 2023] in the paper pdf)  Pre-train, Prompt, and Predict: A Systematic Survey of Prompting Methods in Natural Language Processing - ACM CSUR Jan 2023 https://dl.acm.org/doi/full/10.1145/3560815
>
> [5] (reference [Safavi and Koutra, 2021] in the paper pdf) Relational World Knowledge Representation in Contextual Language Models: A Review - EMNLP 2021
>
> [6] (reference  [Petroni et al., 2019] in the paper pdf) Language Models as Knowledge Bases? - EMNLP 2019
>
> [7] (reference [De Cao et al., 2021] in the paper pdf) Editing Factual Knowledge in Language Models - EMNLP 2021

---

### Official Review · Reviewer_nj8g · 2023-08-04

**Soundness:** 3

**Excitement:**

4: Strong: This paper deepens the understanding of some phenomenon or lowers the barriers to an existing research direction.

**Paper Topic And Main Contributions:**

This survey examines the works that deals with factual knowledge in pre-trained language models, encompassing various methods and datasets designed for factual knowledge probing.

**Reasons To Accept:**

This survey is well-organized  and incorporates numerous relevant studies, reflecting considerable effort and dedication in its compilation. This survey can help people systemically access the recent studies about factual knowledge probing in PLMs.

**Reasons To Reject:**

The insight part of the survey is not very exciting. The future work is too short and hasty.

**Reproducibility:**

N/A: Doesn't apply, since the paper does not include empirical results.

**Reviewer Confidence:**

1: Not my area, or paper was hard for me to understand. My evaluation is just an educated guess.

---

> ### Author Rebuttal · Authors · 2023-08-29
>
> We thank Reviewer nj8g for their feedback and are encouraged that our survey was found well-organized, comprehensive, and beneficial for researchers in this area.
>
> In the following we address the reviewer’s concerns.
>
> ### Survey Insights and Future Work Expansion:
> > The insight part of the survey is not very exciting. The future work is too short and hasty.
>
> We are not quite sure which "insights part" the reviewer is referring to. Section 4 "Insights about Knowledge Retention and Prompt Optimization" discusses insights from our systematically collected papers - and addresses what we know and don't yet know about i) factors affecting knowledge retention and ii) pros and cons of prompt optimization. Section 5 reviews obstacles that currently prevent PLMs from being adopted as KBs. Our future work section was held very brief due to space constraints (the key tables 1 and 2 require 2 pages already). However, with one more page available upon acceptance, we will adapt future work to include the following more detailed and intriguing directions (these are mentioned for Reviewer avt8 as well):
>
> 1. Probing Methods:
>     1. Current probing methods focus mostly on one of the three dimensions (inputs, PLMs, outputs). Simultaneously adapting two or more of these dimensions might lead to further improvements (e.g., optimizing inputs while also debiasing outputs).
> 2. Probing Datasets:
>     1. Some probing datasets are biased towards certain entities [A]. Constructing unbiased probing datasets is essential to have unbiased estimates of factual knowledge in PLMs.
> 3. Knowledge Retention:
>     1. Some architectures and pre-training objectives contribute positively to producing factual outputs [B]. Further developing architectures and training objectives is essential for more progress.
> 4. Prompts Optimization:
>     1. Current methods for discrete prompts optimization lead to better performance, but produce highly un-interpretable prompts [C]. Adapting these methods to produce interpretable prompts would keep the performance gains, and increase the trustworthiness of optimized prompts.
> 5. Obstacles to PLMs as KBs
>     1. Consistency - Multilingual PLMs are sensitive to paraphrased prompts especially in languages other than English [D]. Making PLMs more robust in languages other than English and low-resource languages is crucial for their usability across languages.
>     2. Interpretability - Tracing the factual knowledge of PLMs back to the data it was learnt from is crucial for interpreting the extracted knowledge. This can easily be accomplished for KBs, but remains an open challenge for PLMs. However, the introduction of a fact tracing benchmark [E] opens the door for works in this direction.
>     3. Updating Knowledge - When updating facts in PLMs, it is essential that only the targeted facts are affected and that these facts are retrievable using different paraphrases [F]. Current methods still do not fulfill these requirements.
> 6. Evaluation:
>     1. Currently, factual knowledge in PLMs is mostly evaluated based on how often can PLMs provide correct predictions to a set of inputs. This kind of evaluation neglects many other crucial factors (the number and frequency of the answers in the pre-training corpus, creation period of the pre-training corpus, size of the model, number of training epochs, etc). We call for a more comprehensive evaluation that takes all of these factors into account.
>
> [A]: Knowledgeable or Educated Guess? Revisiting Language Models as Knowledge Bases (Cao et al., 2021)
>
> [B]: REALM: Retrieval-Augmented Language Model Pre-training (Guu et al., 2020)
>
> [C]: Factual Probing Is [MASK]: Learning vs. Learning to Recall (Zhong et al., 2021)
>
> [D]: Factual Consistency of Multilingual Pretrained Language Models (Fierro & Søgaard, 2022)
>
> [E]: Towards Tracing Knowledge in Language Models Back to the Training Data (Akyurek et al., 2022)
>
> [F]: Editing Factual Knowledge in Language Models (De Cao et al., 2021)

---

### Official Review · Reviewer_avt8 · 2023-08-04

**Soundness:** 3

**Excitement:**

3: Ambivalent: It has merits (e.g., it reports state-of-the-art results, the idea is nice), but there are key weaknesses (e.g., it describes incremental work), and it can significantly benefit from another round of revision. However, I won't object to accepting it if my co-reviewers champion it.

**Paper Topic And Main Contributions:**

*  This paper provide a comprehensive review on factual knowledge probing in Pre-trained language models. They categorize relevant studies based on probing inputs, outputs and target LMs.
*  Moreover, this paper provides interesting discussion about knowledge retention, prompt optimization and obstacles of PLMs-as-KBs.


---

After rebuttal: thanks for providing such a comprehensive response.
Overall, I appreciate the author's meticulous review and summarization of the entire field.

Regarding the response about differences between other surveys:
> In contrast to all existing surveys, we conduct a systematic analysis of methods for factual knowledge probing, providing a categorization scheme (that is admittedly novel and well-organized). We additionally provide a categorization of factual probing datasets. Furthermore, we discuss recent findings on knowledge retention, the use of optimized prompts, and challenges with corresponding recent solutions to adopting PLMs as KBs, shedding light on several future work directions.

I appreciate the categorization of relevant studies and probing datasets.
However, I still think that this review lacks novel insights: knowledge retention (the conclusion seems to be obvious), the use of optimized prompts can be found in [3], the discussions about LMs-as-KBs can be found in both [1][2][3].

Overall, I'm willing to increase the score on "Excitement," **but I strongly recommend that the author incorporate more novel discussions in the revision**.

**Questions For The Authors:**

* Can you describe in detail the main differences and novelty of this review and the other reviews mentioned above?

**Reasons To Accept:**

* The paper offers a comprehensive review of factual knowledge probing, the paper is well-written and easy to follow.
*  The categorization schema is novel and well-structured, providing a clear approach to organizing relevant studies.
* The article provides a comprehensive summary of the existing probing methods and datasets (as presented in Table 1 and Table 2).

**Reasons To Reject:**

* Regarding the novelty of the study: many discussions can be found in previous reviews with similar arguments, especially in the most intriguing parts, Section 4 and Section 5, where most of the viewpoints (prompt optimization , consistency, knowledge updating, Interpretability, etc.) have already been summarized in other surveys [1][2][3][4]...
* For a comprehensive review, the discussion on future directions and conclusions appears to be too brief, limiting the ability of future researchers to gain deeper insights.

[1] Language Models As or For Knowledge Bases

[2] A Review on Language Models as Knowledge Bases

[3] The Life Cycle of Knowledge in Big Language Models: A Survey

[4] Pre-train, Prompt, and Predict: A Systematic Survey of Prompting Methods in Natural Language Processing

**Reproducibility:**

5: Could easily reproduce the results.

**Reviewer Confidence:**

4: Quite sure. I tried to check the important points carefully. It's unlikely, though conceivable, that I missed something that should affect my ratings.

---

> ### Author Rebuttal · Authors · 2023-08-29
>
> We thank Reviewer avt8 for their feedback and are encouraged that our survey was found easy to follow, with a helpful categorization scheme to organize the comprehensive set of papers.
>
> In the following we address the two concerns. The first one is closely related to the question for the authors, thus we combine them.
>
> ### How our work differs from other surveys:
>
> > many discussions can be found in previous reviews with similar arguments, especially in the most intriguing parts, Section 4 and Section 5, where most of the viewpoints (prompt optimization , consistency, knowledge updating, Interpretability, etc.) have already been summarized in other surveys [1][2][3][4]...
>
> > Can you describe in detail the main differences and novelty of this review and the other reviews mentioned above?
>
> The reviewer mentioned that our work is similar to other related works. We addressed the relation to other surveys in the paper pdf (L82-86 in the Introduction), limiting ourselves to two works (including [4]). We provide an extension to this part of our work (relation to other surveys) below, as we would include it in a CR version. Additionally, we detail the differences between our work and the 4 works mentioned by the reviewer below. We would also gladly discuss these references in detail in the CR version if the reviewer deems it beneficial.
>
> **Relation to the Surveys (paper version):**
>
> AlKhamissi et al. [2] elaborate requirements for PLMs as knowledge bases and review recent literature w.r.t. those requirements. These requirements are widely known (e.g., consistency [6] and updating knowledge [7]). Our analysis leads to similar general observations (cf. Section 5), and additionally provides more recent solutions to these obstacles (cf. Section 5 and  “Details on Relation to [2]”). Cao et al. [3] cover probing PLMs as part of the knowledge cycle in PLMs, but do not address **factual** knowledge probing at the same level of detail as we do. Liu et al. [4] survey prompting methods in detail. However, this covers only a part of factual knowledge probing methods. Safavi and Koutra [5] survey how PLMs acquire relational knowledge, organizing knowledge representations strategies in PLMs based on different levels of KBs supervision. In contrast to all existing surveys, we conduct a systematic analysis of methods for factual knowledge probing, providing a categorization scheme (that is admittedly novel and well-organized). We additionally provide a categorization of factual probing datasets. Furthermore, we discuss recent findings on knowledge retention, the use of optimized prompts, and challenges with corresponding recent solutions to adopting PLMs as KBs, shedding light on several future work directions. In contrast to other works [2,3,4,5], we employed a systematic approach to curate and analyze relevant literature. This ensured a comprehensive and unbiased representation of existing works.
>
>
> **Details on Relation to [1]:**
>
> [1] is a position paper arguing that PLMs cannot replace KBs. In our work, the potential use of PLMs as KBs is part of the motivation for the survey. In contrast to [1], we do not argue for or against PLMs as KBs, but rather focus on the tool (probing) that would help the community decide in which direction to go. We do cover obstacles to adopting PLMs as KBs in Section 5, in order to review recent solutions that address these obstacles, and to provide directions for future work. In contrast to us, [1] does not cover probing methods and datasets, knowledge retention, and prompts optimization. Even with respect to PLMs-as-KBs, recent solutions for the main challenges (Section 5) are not covered by [1]. To illustrate this on the topic of “Provenance”, [1] states that “LMs have no ability to trace their outputs back to specific source documents (and passages) in the training data”. However, recent work (Towards Tracing Knowledge in Language Models Back to the Training Data (Akyurek et al., 2022)) has introduced a fact tracing benchmark analyzing this issue, laying the groundwork for future research in this direction, and showing that it is possible, albeit challenging, to attribute generated knowledge to training data.
>
> **Details on Relation to [2]:**
>
> [2] focuses on PLMs as KBs, and which requirements PLMs should fulfill to become alternatives for KBs. It does not cover probing methods at the same level of detail and depth as we do, and does not provide any categorization for these as we do (see Table 1). Probing datasets (see Table 2) are also not included in [2], and it does not discuss knowledge retention and prompts optimization as we do. [2] intersects with Section 5 in our work, where we review challenges and solutions to PLMs-as-KBs. Compared to [2] we cover more recent solutions to the identified problems in a dynamic and rapidly changing area of research (e.g., Towards Tracing Knowledge in Language Models Back to the Training Data (Akyurek et al., 2022), Knowledge Neurons in Pretrained Transformers (Dai et al., 2022), Plug-and-Play Adaptation for continuously-updated QA (Lee et al., 2022)).
>
> **Details on Relation to [3]:**
>
> [3] reviews the life cycle of knowledge in PLMs, in which knowledge probing is a part. While [3] is more general and considers knowledge probing in **general**, we focus on **factual** knowledge probing. As a result, our work is more comprehensive and in-depth with respect to factual knowledge probing. To illustrate this, [3] mentions only 4 factual probing datasets (see Table 1 in [3]), whereas our work provides a categorization of more than 35 datasets (see Table 2). Furthermore, some recent important works about knowledge probing are missing  in [3] (e.g., Rewire-then-Probe: A Contrastive Recipe for Probing Biomedical Knowledge of Pre-trained Language Models (Meng et al.,2022), PromptGen: Automatically Generate Prompts using Generative Models (Zhang et al., 2022), Entity Cloze By Date: What LMs Know About Unseen Entities (Onoe et al., 2022)).
>
>
> **Details on Relation to [4]:**
>
> We briefly mentioned the relation to [4] in the Introduction L84-85, and provide more details here: [4] reviews prompting methods in general without a structured literature review. Our work systematically selects and analyzes methods for factual knowledge probing, which include but are not limited to prompting methods. For example, probing datasets are not included in [4], and it does not discuss knowledge retention and recent solutions to LMs-as-KBs. Besides, [4] does not cover works that adapt PLMs (e.g., How Much Knowledge Can You Pack Into the Parameters of a Language Model? (Roberts et al., 2020), Meta-learning via Language Model In-context Tuning (Chen et al., 2022)).
>
>
> [1] Language Models As or For Knowledge Bases - Workshop on Deep Learning for Knowledge Graphs 2022  nonarxiv version https://ceur-ws.org/Vol-3034/paper2.pdf
>
> [2] A Review on Language Models as Knowledge Bases, arXiv version April 2022 - according
> to the first authors web-page ((https://bkhmsi.github.io)  this is under review at JAIR
>
> [3] The Life Cycle of Knowledge in Big Language Models: A Survey, arXiv version March 2023 - this might be under submission or accepted and not yet published, the first author's homepage (https://c-box.github.io) mentions "Machine Intelligence Research" for this paper
>
> [4] (reference [Liu et al., 2023] in the paper pdf)  Pre-train, Prompt, and Predict: A Systematic Survey of Prompting Methods in Natural Language Processing - ACM CSUR Jan 2023 https://dl.acm.org/doi/full/10.1145/3560815
>
> [5] (reference [Safavi and Koutra, 2021] in the paper pdf) Relational World Knowledge Representation in Contextual Language Models: A Review - EMNLP 2021
>
> [6] (reference  [Petroni et al., 2019] in the paper pdf) Language Models as Knowledge Bases? - EMNLP 2019
>
> [7] (reference [De Cao et al., 2021] in the paper pdf) Editing Factual Knowledge in Language Models - EMNLP 2021
>
>
> ### Extending Future Work:
> > For a comprehensive review, the discussion on future directions and conclusions appears to be too brief, limiting the ability of future researchers to gain deeper insights.
>
> Our short future work section is attributed to the space constraints (the - in our view - very relevant tables 1 and 2 require 2 pages in total already). While we covered the most relevant and broadly acknowledged directions, there are more intriguing avenues for future work. With the additional space allowed upon acceptance, we will include discussions on the following further directions:
> 1. Probing Methods:
>     1. Current probing methods focus mostly on one of the three dimensions (inputs, PLMs, outputs). Simultaneously adapting two or more of these dimensions might lead to further improvements (e.g., optimizing inputs while also debiasing outputs).
> 2. Probing Datasets:
>     1. Some probing datasets are biased towards certain entities [A]. Constructing unbiased probing datasets is essential to have unbiased estimates of factual knowledge in PLMs.
> 3. Knowledge Retention:
>     1. Some architectures and pre-training objectives contribute positively to producing factual outputs [B]. Further developing architectures and training objectives is essential for more progress.
> 4. Prompts Optimization:
>     1. Current methods for discrete prompts optimization lead to better performance, but produce highly un-interpretable prompts [C]. Adapting these methods to produce interpretable prompts would keep the performance gains, and increase the trustworthiness of optimized prompts.
> 5. Obstacles to PLMs as KBs
>     1. Consistency - Multilingual PLMs are sensitive to paraphrased prompts especially in languages other than English [D]. Making PLMs more robust in languages other than English and low-resource languages is crucial for their usability across languages.
>     2. Interpretability - Tracing the factual knowledge of PLMs back to the data it was learnt from is crucial for interpreting the extracted knowledge. This can easily be accomplished for KBs, but remains an open challenge for PLMs. However, the introduction of a fact tracing benchmark [E] opens the door for works in this direction.
>     3. Updating Knowledge - When updating facts in PLMs, it is essential that only the targeted facts are affected and that these facts are retrievable using different paraphrases [F]. Current methods still do not fulfill these requirements.
> 6. Evaluation:
>     1. Currently, factual knowledge in PLMs is mostly evaluated based on how often can PLMs provide correct predictions to a set of inputs. This kind of evaluation neglects many other crucial factors (the number and frequency of the answers in the pre-training corpus, creation period of the pre-training corpus, size of the model, number of training epochs, etc). We call for a more comprehensive evaluation that takes all of these factors into account.
>
> [A]: Knowledgeable or Educated Guess? Revisiting Language Models as Knowledge Bases (Cao et al., 2021)
>
> [B]: REALM: Retrieval-Augmented Language Model Pre-training (Guu et al., 2020)
>
> [C]: Factual Probing Is [MASK]: Learning vs. Learning to Recall (Zhong et al., 2021)
>
> [D]: Factual Consistency of Multilingual Pretrained Language Models (Fierro & Søgaard, 2022)
>
> [E]: Towards Tracing Knowledge in Language Models Back to the Training Data (Akyurek et al., 2022)
>
> [F]: Editing Factual Knowledge in Language Models (De Cao et al., 2021)

---

### Meta-Review · Area_Chair_Dfj4 · 2023-09-17

**Recommendation:** 2

**Metareview:**

This is a survey paper on factual knowledge probing for pretrained language models. It categorizes and reviews probing methods and datasets. It also discusses knowledge retention and prompt optimization in further detail.

The paper is fairly comprehensive in reviewing past work in an important and well-studied area. However, reviewers have pointed out that it is quite similar to other recent surveys. There are indeed differences between this paper and each of the individual previous papers especially in terms of scope and depth. However, this paper provides incremental insights over recent surveys taken together.

---

### Decision · Program_Chairs · 2023-10-07

**Decision:**

Accept-Findings

**Comment:**

This is a survey paper on factual knowledge probing for pretrained language models. It categorizes and reviews probing methods and datasets. It also discusses knowledge retention and prompt optimization in further detail.

The paper is fairly comprehensive in reviewing past work in an important and well-studied area. However, reviewers have pointed out that it is quite similar to other recent surveys. There are indeed differences between this paper and each of the individual previous papers especially in terms of scope and depth. However, this paper provides incremental insights over recent surveys taken together.